# Dual-Branch Fusion with Style Modulation for Cross-Domain Few-Shot Semantic Segmentation

## ABSTRACT

Cross-Domain Few-Shot Semantic Segmentation (CD-FSS) aims to achieve pixel-level segmentation of novel categories across various domains by transferring knowledge from the source domain leveraging limited samples. The main challenge in CD-FSS is bridging the inter-domain gap and addressing the scarcity of labeled samples in the target domain to enhance both generalization and discriminative abilities. Current methods usually resort to additional networks and complex strategy to embrace domain variability, which inevitably increases the training costs. This paper proposes a Dual-Branch Fusion with Style Modulation (DFSM) method to tackle this issues. We specifically deploy a parameter-free Grouped Style Modulation (GSM) layer that captures and adjusts a wide spectrum of potential feature distribution changes, thus improving the model's solution efficiency. Additionally, to overcome data limitations and enhance adaptability in the target domain, we develop a Dual-Branch Fusion (DBF) strategy which achieves accurate pixel-level prediction results by combining predicted probability maps through weighted fusion, thereby enhancing the discriminative ability of the model. We evaluate the proposed method on multiple widely-used benchmark datasets, including FSS-1000, ISIC, Chest X-Ray, and Deepglobe, and demonstrate superior performance compared to state-of-the-art methods in CD-FSS tasks.

## CCS CONCEPTS

• **Computing methodologies** → **Image representations**.

## KEYWORDS

cross-domain, few-shot semantic segmentation, style modulation, adaptive fusion

## 1 INTRODUCTION

Deep learning-powered semantic segmentation methods achieve outstanding performance, which heavily relies on a substantial amount of pixel-wise labeled data. However, this reliance can restrict their ability to learn effectively from a few samples in complex scenarios and to generalize to new categories. Few-shot semantic segmentation (FSS) is proposed to train a model with excellent capability for both generalization and adaptability when confronted with previously unseen classes. The core of this task is to rapidly learn new concepts and effectively segment images from new categories using minimal annotated data. Despite the significant advancements in the FSS task, the current methods primarily focus on the intra-domain scenario, where the training and testing data belong to the same domain. However, this assumption often falls short of real-world requirements, where the differences between the training and testing data arise due to variations originating from diverse sources or changes over time. As a result, the training sets become outdated, which can significantly affect the performance of the FSS task. CD-FSS has attracted interest as a solution to circumvent these restrictions. It enables the utilization of data from different domains and label spaces to be employed as source and target data, respectively.

As shown in Figure 1, CD-FSS tasks involve training a model on a source domain with abundant pixel-level annotated data like PASCAL VOC [9] and applying it to a target domain with a completely different data distribution and categories, which has only a limited number of labeled samples like a skin lesion dataset [6]. Therefore, the primary challenges are the disparities in data distribution between the source and target domains, as well as the scarcity of labeled samples in the target domain. The former results in limited inter-domain correlation, thereby reducing the effectiveness of knowledge transfer. The latter leads to inadequate feature extraction within the domain, which in turn weakens the learning performance of class-specific semantic knowledge. As described, our primary focus is on addressing two key issues: 1) How to minimize the visual differences between the training and testing domains to improve the model's generalizability; 2) How to predict unknown categories in the face of limited samples, thus enhancing the model's discriminative ability.

Recently, many methods for CD-FSS tasks have been studied, which can be divided into two main categories. The first category of works [3, 14] is focused on the objective of training a model with robust generalization on the source domain while lacking access to target domain data. These methods enhance the model's generalizability by constructing complex parameter modules. However, the substantial domain gap between the source and target domains makes it difficult to achieve effective model transfer, leading to unsatisfactory results. Therefore, to achieve better segmentation outcomes, Lei et al. [17] introduce target domain data during the fine-tuning stage to learn domain-specific features, i.e., domain adaptation. Considering the limited labeled data in the target domain, we advocate for this direction.

In order to tackle the aforementioned problems, we propose a novel CD-FSS approach named Dual-Branch Fusion with Style Modulation (DFSM) to enhance the model's generalizability and improve its discriminative capabilities in the target domain. Specially, to handle the first challenge, we introduce a Grouped Style Modulation (GSM) layer to generate diverse domain styles, thereby enhancing the model's generalization capabilities across different

**Unpublished working draft. Not for distribution.**

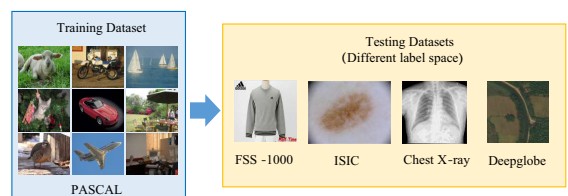

**Figure 1: In CD-FSS tasks, the training and testing sets come from distinct domains, resulting in significant disparities in their data distributions. Additionally, the label spaces of the training and test sets do not intersect.**

domains. Inspired by [13, 19], we significantly alter the domain style by adjusting the statistical properties of deep network features, specifically by transforming the mean and variance of features to those of the target domain. However, during the training process, the target domain remains unknown, making it challenging to identify an appropriate target domain style. Existing methods attempt to simulate the target style by swapping styles of source domain data or using learnable style parameters, but these approaches increase the complexity of the model and extend the training duration. To address these challenges, we adopt a parameter-free approach, capturing the diversity of the source domain from an implicit semantic enhancement perspective within the feature space, thus broadening the coverage for unknown target domains. Based on statistical analysis of source domain data, we construct latent style estimates to capture the potential directions and magnitudes of style changes between domains. In addition, we leverage channel grouping strategies to extend the spectrum of possible style representations.

To address the second challenge, we devise a Dual-Branch Fusion (DBF) strategy to enhance the model's discriminatory ability and domain adaptability through efficient utilization of available data and knowledge transfer. On the one hand, considering the limitation of sample quantity in the target domain, we employ data augmentation techniques to expand the sample space, thus increasing the diversity and robustness of model fine-tuning. By treating augmented support samples as pseudo-support samples and original support samples as pseudo-query samples, we provide additional training signals for fine-tuning, further improving the model's performance in the target domain. On the other hand, to enhance the model's discriminative power in specific domains, we propose not merely updating the parameters of the pre-trained model to adapt to the new domain but instead construct a dual-branch model to augment decision support information, delivering precise pixel-level predictions. By adaptively integrating predicted probability maps via weighted fusion, our model not only resolves the issue of scarce samples in the target domain but also enhances the model's ability to discriminate.

The primary contributions can be described as follows:

- We propose a Dual-Branch Fusion with Style Modulation (DFSM) method for CD-FSS that effectively bridges the domain gap, thereby improving both generalization and discrimination in specific domains.

- We introduce a parameter-free Grouped Style Modulation (GSM) layer to generate diverse domain styles aimed at enhancing the model's generalizability. Our latent style estimation effectively captures the appropriate range of potential feature distribution variations.
- We develop a Dual-Branch Fusion (DBF) strategy to elevate the model's discriminative power and domain adaptability in the target domain by expanding the sample space and adaptively integrating predicted probability maps via weighted fusion.
- We conduct extensive experiments on four widely used benchmark datasets. The results indicate that our method significantly improves upon the baseline models and outperforms the state-of-the-art CD-FSS algorithm.

## 2 RELATED WORK

### 2.1 Cross-domain Semantic Segmentation

The current researchs on cross-domain semantic segmentation can be categorized into two groups: domain adaptation (DA) and domain generalization (DG). In the context of domain adaptation tasks, a model that has been trained effectively on the source domain must undergo fine-tuning using test data prior to its deployment in the target domain. A variety of [5, 15, 26, 33] concentrate on adversarial alignment between the source and target domains, unsupervised learning using pseudo-labels in the target domain, as well as combining adversarial adaptation with self-training or pixel-level adaptation. Unlike domain adaptation, in the training phase of domain generalization tasks, the model lacks access to data from the target domain. Existing domain generalization methods [8, 21, 24, 34] can be categorized into domain invariance and feature decoupling based on representation learning. The former achieves its goal through invariant risk minimization, kernel-based approaches, explicit feature alignment, and domain adversarial learning. The latter decomposes features into domain-shared and specific-domain components to facilitate better generalization capability of the model.

### 2.2 Few-shot Segmentation

The metric mechanism plays a vital role in the study of few-shot semantic segmentation, based on the meta-learning paradigm. Previous research can be classified into two main categories: prototype-based approaches and parameterized-based methods, regardless of whether the metric approach involves trainable parameters or not. Prototype-based methods [4, 18, 22] extract representative features or prototypes from a limited set of training samples and classify new samples to be segmented by calculating their distance relationships with the prototypes of each category, achieving precise pixel-level segmentation. Parameterized-based methods [16, 23, 25, 32] typically consist of encoders, feature processors, and decoders. During training, such methods dynamically adjust the network's parameter structure to learn the similarities between samples adaptively. However, these methods solely focus on learning performance within analogous data settings. In practical applications, they often face inconsistent data distributions across various domains. Finite data resources constrain these methods, making it difficult to effectively enhance generalization capabilities in new domains.

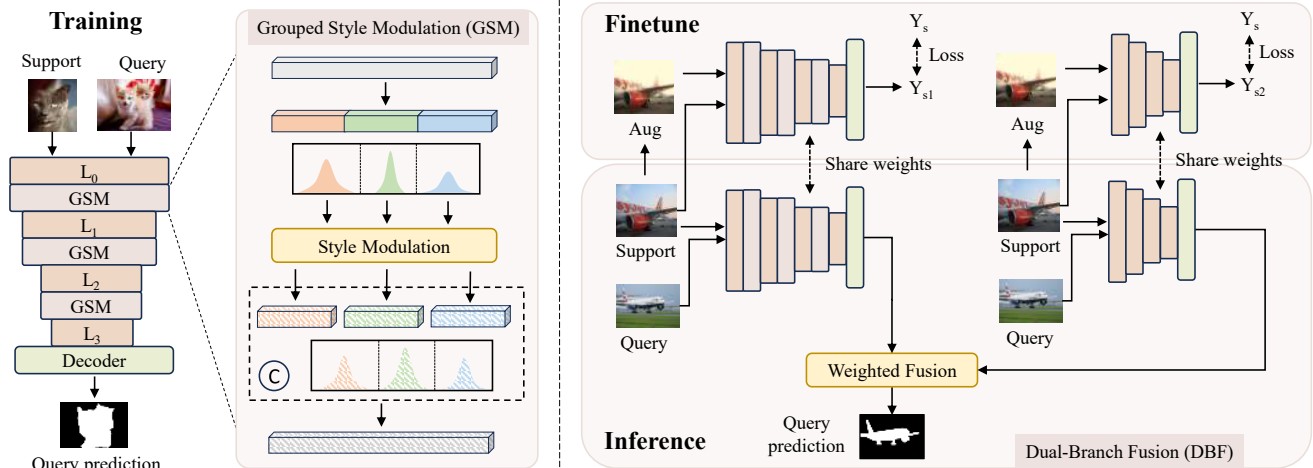

**Figure 2: Illustration of the proposed method. It consists of two phases: 1)Training phase. We incorporate a GSM after each layer of the feature extractor to generate diverse feature representations, enhancing the model's generalizability. It is worth noting that GSM does not introduce any learnable parameters, thereby reducing the computational complexity of the model. 2)Fine-tuning and inference phase. We establish a dual-stream branch to learn effective feature representations with each branch is fine-tuned independently. Subsequently, during the inference stage, we perform a weighted fusion of the predictions from the dual streams.**

## 2.3 Cross-domain Few-shot Semantic Segmentation.

Compared with cross-domain semantic segmentation and FSS, the training set and testing set in CD-FSS exhibit differences in data distribution and no label distribution overlap. To alleviate inter-domain discrepancies, Wang et al. [28] develop a meta-memory that collects statistical characteristics of instances from the source domain, thereby reducing the domain gap between the source and target domains. They also adopt contrastive learning to regulate the relationships among prototypes of different novel categories. Lei et al. [17] propose a model called PATNet, which utilizes a pyramid module to convert the features acquired from the source domain into features that are invariant across different domains. However, mapping features across domains may only sometimes be accurate, which can result in misaligning well-aligned support-query pairs in the original space when transferred to the new space. To address this issue, Huang et al. [14] propose the RestNet based on PAT-Net, which utilizes residual connections to bridge the gap between the original space and the latent space. However, these methods require additional parameters. Our approach learns diversified domain styles through a parameter-free GSM layer to reduce model complexity.

## 3 METHOD

### 3.1 Preliminaries

In the Cross-Domain Few-Shot Semantic Segmentation task, there exists a source domain $D_s = (X_s, Y_s)$ and a target domain $D_t = (X_t, Y_t)$, where $X, Y$ denote the data distributions and label spaces respectively. There is a significant domain shift between the source

and target domains, and their label spaces have no intersection, i.e., $X_s \cap X_t = \varnothing, Y_s \cap Y_t = \varnothing$. The CD-FSS task leverages a pre-trained model from the source domain to divide new class samples in the target domain. The training set $D_{train}$ and test set $D_{test}$ are respectively composed of $D_s = (X_s, Y_s)$ and $D_t = (X_t, Y_t)$. The same episode mechanism used in few-shot semantic segmentation is adopted throughout the training and evaluation process, meaning that each training or testing scenario instantiates a specific instance of a segmentation learning task of $C - way, K - shot$. Specifically, both training and evaluation involve multiple scenarios, each of which consists of a support set $S_i = \{(I_s^{c,k}, M_s^{c,k})\}_{k=1}^{K}$ and a query set $Q_i = \{(I_q^{c,k}, M_q^{c,k})\}_{k=1}^{K}$, where the support set contains $K$ support images for evaluating $C$ semantic categories, i.e., $c \in C_i, | C_i |= C$, with $I, M$ representing an image and a mask respectively.

To prepare for training the CD-FSS model, start by selecting $C$ classes at random from the corresponding dataset. Then, pick $K$ samples for each selected class to create a support set. Additionally, choose $N_{query}$ samples from each class to form a query set. It is crucial to guarantee that there is no intersection between the support and query sets within the same situation. The CD-FSS model is trained on the training set $D_{train}$ of the source domain during the training process. The model is not allowed to access any data from the target domain at this stage. Finally, the well-trained model is evaluated on the test set $D_{test}$ of the target domain in testing phase.

### 3.2 Architecture Overview

**Design of the framework.** The overall framework of the proposed method is illustrated in Figure 2, using a feature extractor with the

first three layers of ResNet-50 [12] as the backbone network to generate the initial visual feature for each sample. To explore potential domain styles, a group style modulation layer is strategically positioned after each layer. We then use the decoder from SSPNet [10] that refines and fuses prototypes from support and query samples to make accurate predictions on the query sample.

**Two-stage Methods** As shown in Figure 2, our method consists of two learning stages. In the first stage, the model is trained on the source domain. In the second stage, the model is fine-tuned on the target domain. Specially, the model parameters obtained during the training phase will serve as the initial parameters for the dual-branch model during the fine-tuning stage. Our objective is to fine-tune the dual branches independently to adapt to the specific domain's style, and then fuse the predictions of both branches during the inference stage.

## 3.3 Grouped Style Modulation

Effective simulation of diverse style information within the training domain is crucial to enhance generalization capability. In this work, we propose a Grouped Style Modulation (GSM) to predict potential style variations between domains, thereby enhancing the model's robustness across different testing tasks. Existing methods often generate diverse styles through deep learning modules or random strategies, increasing computational costs with parameter-based learning and potentially producing meaningless styles that degrade segmentation performance. Inspired by previous works [13, 19], we introduce a GSM that capture potential style estimates from small batches of data, providing each feature channel with an appropriate and meaningful range of variation. This approach does not compromise model training but enables the simulation of different potential shifts.

**Statistics Calculation**. The statistics of features, such as mean and standard deviation, can represent the style of an image. Therefore, our initial step involves acquiring the feature statistics from both support and query samples. For each iteration, the intermediate feature $F^l \in \mathbb{R}^{B \times C \times H \times W}$ is extracted by the feature extractor for each sample from current batch of training data $\mathcal{B} = \{(I_s, I_q)\}_{i=1}^{B}$, where $I_s \in \mathbb{R}^{3 \times H_s \times W_s}$ and $I_q \in \mathbb{R}^{3 \times H_q \times W_q}$ represent support set and query set respectively. Then we compute the channel-wise feature statistic including mean $\mu(\cdot)$ and standard deviation $\sigma(\cdot)$ of given feature via:

$$\mu(F^l) = \frac{1}{HW} \sum_{h=1}^{H} \sum_{w=1}^{W} F^l \quad (1)$$

$$\sigma(F^l) = \sqrt{\frac{1}{HW} \sum_{h=1}^{H} \sum_{w=1}^{W} (F^l - \mu(F))^2 + \epsilon} \quad (2)$$

where $B, C, H, W$ represent batch size, channel, height and weigth of $F^l$, respectively. $\epsilon$ is set to 1e-6. By applying equation 10 and equation 2 to $\mathcal{B}$, we can obtain the mean $\mathcal{U}^l \in \mathbb{R}^{B \times C}$ and standard deviation $\mathcal{S}^l \in \mathbb{R}^{B \times C}$ of all $B$ samples in the $l$-th intermediate layer. Subsequently, we obtain a representation of the domain style by calculating the average of all feature statistics ($\mathcal{U}^l$ and $\mathcal{S}^l$) within the batch, namely, the domain mean $u^l$ and standard deviation $s^l$.

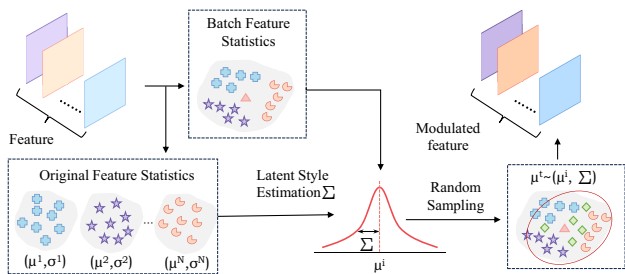

**Figure 3: Illustration of the Style Modulation approach which modulates domain styles without introducing additional parameters.**

**Latent Style Estimation**. Assuming that the distribution of the target feature statistics follows a multivariate Gaussian distribution, the center of this distribution can be considered as the original feature statistics for each feature, while the standard deviation describes the range of different potential offsets. To obtain an appropriate and meaningful range of variation, we characterizes latent style estimation from a semantic enhancement perspective. There exist multiple distinct semantic directions within the deep feature space, and by translating features along specific directions and magnitudes, semantic transformations of the features can be realized. The variance among features encapsulates inherent semantic information. Consequently, we utilizes the variance of feature statistics across all batches of samples to construct latent style estimation.

$$\sum_{\mu}^{l} = \sqrt{\frac{1}{B} \sum_{b=1}^{B} (\mathcal{U}_b^l - u^l)^2} \quad (3)$$

$$\sum_{\sigma}^{l} = \sqrt{\frac{1}{B} \sum_{b=1}^{B} (\mathcal{S}_b^l - s^l)^2} \quad (4)$$

where, $\sum_{\mu}^{l}, \sum_{\sigma}^{l}$ represent latent style estimation of domain mean and variance. It is worth noting that the magnitude of latent style estimation plays a critical role in representing potential feature semantic information, providing potential transformations that may occur within style embeddings.

**Grouped Style Modulation**. Once latent style estimations are obtained, corresponding multivariate Gaussian distributions can be constructed to synthesize new domain statistics, i.e., mean $\mu_t \sim N(\mu^l, \sum_{\mu}^{l})$ and variance $\sigma_t \sim N(\sigma^l, \sum_{\sigma}^{l})$, as shown in Figure 3. We sample from the Gaussian distribution and select semantic transformation directions in a random manner for enhancement.

$$\mu_t = \mu^l + \epsilon_\mu \sum_{\mu}^{l} \quad (5)$$

$$\sigma_t = \sigma^l + \epsilon_\sigma \sum_{\sigma}^{l} \quad (6)$$

where $\epsilon_\mu$ and $\epsilon_\sigma$ follows the standard normal distribution $\mathcal{N}(0, 1)$.

Then we utilize affine transformations to transform the style $(\mu, \sigma)$ of the intermediate layer features $F^l$ in the model into a novel

style $(\mu_t, \sigma_t)$ that is randomly sampled:

$$\hat{F}^l = \sigma_t \frac{F^l - \mu^l}{\sigma^l} + \mu_t \tag{7}$$

In practical applications, we group the features of this batch by the channel dimension and perform affine transformations on each group separately. This module can be integrated at various positions within the network. By setting hyperparameters $p$ to control the probability of applying the model, it enhances the model's ability to learn stable features and robustness, thereby improving its generalization capability. Note that this module only operates during training and fine-tuning and can be discarded during testing. It is worth noting that this module does not introduce any learnable parameters, which to some extent reduces the computational complexity of the model. By introducing GSM, we can capture the variability of feature distributions, aiding the model in learning a more diverse range of domain styles. Compared to traditional methods, this strategy, which does not require additional complex calculations, not only improves the model's generalization but also reduces the difficulty and computational cost of model training.

### 3.4 Dual-Branch Fusion.

To further enhance the model's discriminative ability in specific domains, we employ a dual-branch fusion strategy to fine-tune the model, aiming to ensure that it adapts to the data distribution of the target domain.

**Dual-Branch Joint Modeling.** In the target domain, support samples and query samples representing the same category may exhibit different appearances. The limited number of support samples restricts the model's ability to learn about the current category, thereby reducing its predictive performance on the query samples. Introducing a Group Style Modulation (GSM) layer during the fine-tuning phase can enhance the model's generalizability. However, due to the limitations of few samples in CD-FSS, the model may produce ambiguous predictions in some complex areas, leading to reduced discriminative power. Under limited data conditions, we employ an auxiliary model to augment decision support information, thereby enhancing the model's discriminatory ability.

Therefore, during the fine-tuning and testing phase, we design two parallel branches to enhance the model's discriminative ability for the current task. Specifically, the first branch consists of two components: a feature extractor and a decoder formed by a prototypical network. Similar to the training phase, we use ResNet-50 as the feature extractor and add a GSM layer after each residual block. The second branch is based on the first but with the GSM layer removed.

**Weighted Fusion.** As shown in Figure 4, the branch with GSM captures excessive class-irrelevant information, the branch without GSM loses some class-relevant information, while the fusion method achieves accurate pixel-level prediction results. In this work, we further utilize the predictive differences between the two branches as an uncertainty estimation. Specifically, during the inference stage, after obtaining predicted probability maps from both branches for a query sample, we adopt a weighted fusion approach to integrate the prediction results. Formally, for a given query sample $F_q \in \mathbb{R}^{B \times C \times H \times W}$ and the predicted mask $M_q \in \mathbb{R}^{B \times 2 \times H \times W}$, we first perform mask average pooling to obtain the foreground

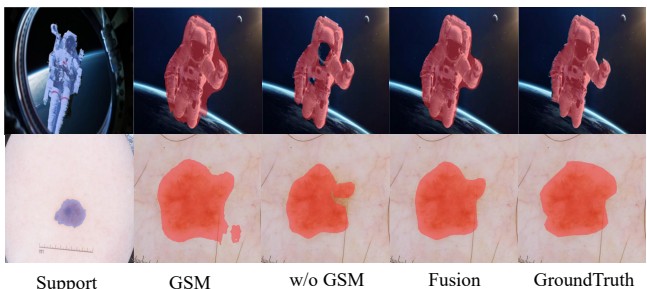

| Support | GSM | w/o GSM | Fusion | GroundTruth |

**Figure 4: Illustration of the visual prediction results on query samples using branches with GSM, branches without GSM, and the method that uses dual-branch fusion.**

prototype $FP_q \in \mathbb{R}^{B \times C}$ and background prototype $BP_q \in \mathbb{R}^{B \times C}$. Subsequently, cosine similarity is used to compute the similarity between the foreground and background.

$$S(FP_q, BP_q) = \frac{1}{BC} \sum_{b,c} cos(FP_q, BP_q) \tag{8}$$

By applying Equation 8 to the prediction results from the two branches, we obtain two similarity scores $S^1, S^2$. Finally, the weight of the first branch can be obtained by:

$$\alpha = \frac{1}{1 + e^{-\frac{S^1 + S^2}{2}}} \tag{9}$$

The fusion of the two branches enhances the overall model's ability to distinguish between different categories within the same domain.

**Weights sharing.** In the fine-tuning and inference stages of each scenario, the two branches share the parameters of the trained model as initial parameters. Additionally, except for the last residual block which requires parameter updates during fine-tuning, the two branches share the same weights. On one hand, sharing weights can reduce the computational complexity during the fine-tuning phase. On the other hand, the knowledge learned by the model in the source domain can be equally transferred to both branches, guiding them to learn quickly.

### 3.5 Network Training and Finetune

We employ a two-stage training approach, where the GSM is used during the training phase to simulate potential domain styles, thereby enhancing the model's adaptability to the target domain. In the fine-tuning phase, data from the target domain is introduced for domain-specific adaptation, allowing the model to conform to the feature distribution of the target domain more accurately.

**Training.** During the training, the fused prototype obtained via a self-matching strategy yields the corresponding prediction results $M_{out}^q$. To facilitate training, new predictions $M_{supp}^s$ and $M_{self}^q$ are derived by predicting the support samples with the support prototype and the query samples with the self-matching prototype, respectively. Subsequently, the cross-entropy loss between the prediction results and the ground truth is calculated. Ultimately, we

**Table 1: The MIoU results for 1-way 1-shot and 5-shot setups in CD-FSS tasks, obtained from few-shot segmentation methods. These methods are all trained on PASCAL VOC.**

| Method | FSS-1000 | | ISIC | | Chest X-ray | | Deepglobe | | Average | |
|---|---|---|---|---|---|---|---|---|---|---|
| | 1-shot | 5-shot | 1-shot | 5-shot | 1-shot | 5-shot | 1-shot | 5-shot | 1-shot | 5-shot |
| PGNet [30] | 62.42 | 62.74 | 21.86 | 21.25 | 33.95 | 27.96 | 10.73 | 12.36 | 32.24 | 31.08 |
| PANet [27] | 69.15 | 71.68 | 25.29 | 33.99 | 57.75 | 69.31 | 36.55 | 45.43 | 47.19 | 55.1 |
| CANet [31] | 70.67 | 72.03 | 25.16 | 28.22 | 28.35 | 28.62 | 22.32 | 23.07 | 36.63 | 37.99 |
| RPMMs [29] | 65.12 | 67.06 | 18.02 | 20.04 | 30.11 | 30.82 | 12.99 | 13.47 | 31.56 | 32.85 |
| PFENet [25] | 70.87 | 70.52 | 23.5 | 23.83 | 27.22 | 27.57 | 16.88 | 18.01 | 34.62 | 34.98 |
| RePRI [1] | 70.96 | 74.23 | 23.27 | 26.23 | 65.08 | 65.48 | 25.03 | 27.41 | 46.09 | 48.34 |
| HSNet [23] | 77.53 | 80.99 | 31.2 | 35.1 | 51.88 | 54.36 | 29.65 | 35.08 | 47.57 | 51.38 |
| PATNet [17] | 78.59 | 81.23 | 41.16 | 53.58 | 66.61 | 70.2 | 37.89 | 42.97 | 56.06 | 61.99 |
| SSPNet [10] | 79.43 | 80.32 | 36.07 | 47.20 | 73.70 | 74.91 | 40.19 | 50.54 | 57.35 | 63.24 |
| PMNet [3] | 84.6 | 86.3 | 51.2 | 54.5 | 70.4 | 74.0 | 37.10 | 41.60 | 60.83 | 64.1 |
| **Ours** | **85.44** | **90.24** | **57.02** | **64.77** | **91.49** | **92.90** | **40.99** | **52.69** | **68.74** | **75.15** |

obtain a weighted loss function:

$$
\begin{aligned}
L_{total} = &\, \alpha_1 BCE(Softmax(M_{out}^q, M_{gt}^q)) + \\
&\, \alpha_2 BCE(Softmax(M_{supp}^s, M_{gt}^s)) + \\
&\, \alpha_3 BCE(Softmax(M_{self}^q, M_{gt}^q))
\end{aligned}
\tag{10}
$$

where $\alpha_1$, $\alpha_2$ and $\alpha_3$ are the balance weights.

**Finetune.** In each episode, we randomly select different query images while constructing augmented samples for the remaining support set, treating it as a pseudo-support set and the original support set as a pseudo-query set. The parameters of the training model are loaded simultaneously for the two branches, and the fine-tuning processes are independent of each other. After training each episode, the support set is immediately used to predict the query set. It is important to note that training gradients at this stage does not involve updates.

## 4 EXPERIMENT

### 4.1 Experimental Setup

**Datasets.** Following previous research, we train our model using the PASCAL [9] dataset as the source domain and evaluate it on target domains including FSS-1000 [20], Deepglobe [7], ISIC2018 [6], and Chest X-ray [2]. The PASCAL dataset is composed of PASCAL VOC 2012 [9] with annotations from the SDS dataset [11]. FSS-1000 [20] consists of 1000 categories split into training, validation, and test sets with 520, 240, and 240 categories, respectively. Each category contains ten images. For evaluation, we selected 2400 support-query pairs from the test set. Deepglobe [7] features satellite images annotated at the pixel level across seven categories. To reduce individual image sizes and expand the test image pool, each image is split into six sections. By removing single-class and unknown area images, we have 5666 images for evaluation. ISIC2018 [6] is comprised of skin lesion images, with each image precisely containing one primary lesion. The dataset includes a total of 2596 images. Chest X-ray [2] contains X-ray images for tuberculosis analysis, collected from 58 cases with tuberculosis manifestations and 80 normal cases, totaling 566 images.

---

**Algorithm 1** Fine-tuning and Inference under 1-shot setting

---

**Require:** Branch1Model $M_{\theta_1}^1$, Branch2Model $M_{\theta_2}^2$, pretrained model $f_\theta$
**Require:** Test episode $T = (S, Q)$
1: Initialize $M_{\theta_1}^1$ and $M_{\theta_2}^2$ with $f_\theta$
2: **while** not done **do**
3:     Generate pseudo-samples $S^{pseudo}$ of support sample $S$
4:     Set support-query pair $T^f = (S^{pseudo}, S)$
5:     Fine-tune the model and update the model parameters $M_{\theta_1}^1$
      using $T^f = (S^{pseudo}, S)$
6: **end while**
7: **while** not done **do**
8:     Generate pseudo-samples $S^{pseudo}$ of support sample $S$
9:     Set support-query pair $T^f = (S^{pseudo}, S)$
10:     Fine-tune the model and update the model parameters $M_{\theta_2}^2$
      using $T^f = (S^{pseudo}, S)$
11: **end while**
12: Test the fine-tuned model using query sample $Q$, $\hat{P^1} = M_{\theta_1}^1(Q), \hat{P^2} = M_{\theta_2}^2(Q)$
13: Compute the prediction results $P$ with weighted fusion

---

**Evaluation metrics.** Existing methods for few-shot semantic segmentation primarily utilize the Mean Intersection over Union (mIoU) as evaluation metrics. The mIoU represents the average IoU across all categories, taking into account different object classes. Higher values of these evaluation metrics indicate better segmentation performance.

**Implementation Details** In this study, we adopt the same experimental setup as PATNet [17]. All samples from the PASCAL-$5^i$ dataset are combined to form the training set, and only one round of experimentation is conducted during the training phase. In the testing phase, to ensure a fair performance comparison with existing methods, we utilize the prevalent evaluation methods currently in use. We will assess all different types of test datasets using the same model, namely, the model trained on the PASCAL-$5^i$ dataset. We employ a pre-trained ResNet-50 on ImageNet as our feature extractor,

utilizing features generated by conv4x. Following the methodology of previous work [17], during the training and fine-tuning phases, we set the spatial dimensions of both support and query images at 400×400. During training, we comprehensively train the model using an SGD optimizer with an initial learning rate of 1e-3, where all parameters of the feature extractor are updated.

## 4.2 Comparison Results

**Quantitative Results.** The algorithm proposed in this paper has been thoroughly compared with the current state-of-the-art methods for cross-domain few-shot semantic segmentation. Specific experimental results are presented in Table 1. All methods are trained on the PASCAL dataset and tested on the FSS, ISIC, Chest X-ray, and Deepglobe datasets, respectively. Bold indicates the best performance among all methods. The results demonstrate that our method achieves the best performance in three out of the four datasets and lead in average metrics. Compared to the second best method PMENet [3], our approach improved accuracy by 7.91% and 11.05% under the 1-shot and 5-shot settings, respectively. These exceptional cross-domain accuracies indicate that our method can effectively generalize to unknown testing domains. As the number of support images increased from one to five, the model performance showed a robust improvement trend. Particularly on the Deepglobe and ISIC datasets, our method exhibited significantly more progress with the additional supervisory information, with performance gains reaching as high as 11.7% and 7.75%, respectively. These results thoroughly validate the efficient learning capability and outstanding generalization performance of our approach.

**Qualitative Results.** Figure 5 illustrates the visualized prediction outcomes of our method on the four datasets (sequentially from top to bottom: FSS, ISIC, Chest X-ray, and Deepglobe datasets). It is noteworthy that, prior to making predictions on the test set, the model had not been exposed to the same categories within the training dataset. The qualitative results imply that with merely 1/5th of a support sample providing categorical target information, the model is still capable of identifying and accurately segmenting the complete foreground object in query images. For instance, the second row of Figure 5 displays the prediction results on skin lesion images. Despite the training data originating from natural images, which significantly differ in style from skin lesion images, the approach employed in this study still manages to closely predict the segmentation boundaries of the primary lesions in the query images.

## 4.3 Ablation Studies

We conducted a series of ablation experiments on the modules proposed in our model. The 1-shot and 5-shot metrics mentioned in the table below represent average values. Specifically, to ensure fairness, evaluations are performed using five different random seeds for each test dataset, and the average of these five trials is taken as the result for that dataset.

**Effectiveness of Specific Modules.** Table 2 shows the impact of different modules on enhancing model performance. We adaptively improve the few-shot semantic segmentation method SSPNet [10], enabling it to effectively handle cross-domain few-shot semantic segmentation tasks. The enhanced SSPNet is then used as a baseline

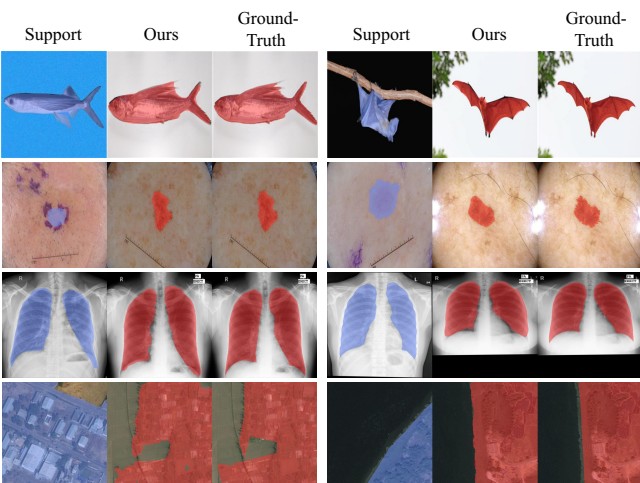

**Figure 5: Prediction results of our model on the 1-shot setting for CD-FSS task. From top to bottom, each row represents FSS-1000, ISIC, Chest X-Ray, and Deepglobe datasets.**

**Table 2: Effectiveness of Specific Modules. FI represents the fine-tuning results using a model with GSM, and DBF indicates the fine-tuning results after employing a dual-branch fusion strategy.**

| Baseline | GSM | FI | DBF | 1-shot | 5-shot |
|:---:|:---:|:---:|:---:|:---:|:---:|
| ✓ | | | | 57.35 | 63.24 |
| ✓ | ✓ | | | 60.26 | 64.98 |
| ✓ | ✓ | ✓ | | 66.08 | 71.81 |
| ✓ | ✓ | ✓ | ✓ | **68.74** | **75.15** |

model for training and evaluation, with the results displayed in the first row of the Table 2. Upon incorporating the GSM, the model's performance increased by 2.91%. This improvement is due to the model's ability to learn more diverse information during the feature extraction phase, thereby boosting its generalization capabilities across various unknown testing scenarios. Additionally, leveraging a fine-tuning strategy allows the model to update parameters according to the visual style of the test domain before the inference stage, thus guiding the inference process for query samples more accurately. This strategy result in a performance increase of 5.82%, confirming the significance of fine-tuning for overall performance enhancement. Furthermore, by computing the weighted output of the dual-branch module, the model is able to achieve a gain of 2.66%. With all modules combined, the method proposed in this study elevates the baseline by 11.39%. These module ablation experiments compellingly demonstrate that the improvements proposed in this research contribute to enhancing the model's generalization ability and category discrimination accuracy on unknown testing domains, effectively boosting the model's overall performance.

**Effectiveness of Grouped Style Modulation Layer.** In Table 2, we clearly observe that the introduction of the GSM leads to significant improvements in model performance under both 1-shot

**Table 3: Effectiveness of Grouped Style Modulation Layer. Ori denotes not adopting a grouping strategy.**

| Group | FSS | ISIC | Chest X-Ray | Deepglobe | Mean |
|-------|-----|------|-------------|-----------|------|
| Ori | 78.05 | 40.16 | 77.26 | 41.41 | 59.22 |
| 8 | 77.98 | **42.06** | 75.73 | 41.01 | 59.20 |
| 16 | **78.20** | 41.50 | **79.32** | **42.01** | **60.26** |
| 32 | 78.19 | 41.44 | 78.56 | 41.12 | 59.83 |

**Table 4: Effectiveness of Dual-Branch Fusion.**

| Method | FSS | ISIC | Chest X-Ray | Deepglobe | Mean |
|--------|-----|------|-------------|-----------|------|
| GSM | 83.45 | 52.22 | 89.58 | 39.05 | 66.08 |
| w/o GSM | 82.23 | 51.31 | 91.31 | 40.86 | 66.43 |
| Mean | **85.7** | 56.34 | 91.39 | **41.28** | 68.68 |
| WF | 85.44 | **57.02** | **91.49** | 40.99 | **68.74** |

and 5-shot settings. In this work, we achieve local style modulation by grouping channels, thereby enhancing the diversity of features. To unveil the impact of channel group size, we conduct ablation studies on the size of the grouped channels. As demonstrated in the table 3, the model achieves the best results when the channel size is 16. Additionally, compared to the model without channel grouping, there is a performance improvement of 1.04%, which underscores the effectiveness of this method.

Moreover, we systematically investigate the potential impact of noise ratio on model performance, where a higher noise ratio corresponds to the generation of a richer variety of domain styles. The results displayed in Figure 6 reveal a trend: as the noise ratio incrementally increases, the model's overall performance shows an upward trajectory. Furthermore, we utilize t-SNE to visualize the feature distribution differences between datasets, as shown in Figure 7. The pink dots are features extracted from the training domain dataset PASCAL, while the orange dots represent features extracted from the test domain dataset Deepglobe, using the feature extractor. The left image depicts the feature distribution between the two datasets under the baseline model. It is evident that there is a significant domain discrepancy between the two datasets. However, after employing the GSM proposed in this study, the distribution between the two datasets shifes from a clustered aggregation to a radial, point-like spread, significantly reducing the distribution gap, as shown in the right image. This illustrates the effectiveness of the GSM.

**Effectiveness of Dual-Branch Fusion.** To verify the effectiveness of DBWF, wo conduct some comparative experiments as shown in Table 4. In the experiment named w/o GSM, we remove the operation of GSM. We discover that each branch exhibits distinct performance advantages on different datasets. Therefore, we further consider combining the two aforementioned methods. The method called Mean integrates the two methods equally and achieves a performance gain, which demonstrates the necessity of complementarity in prediction results. Moreover, utilizing weighted fusion can help the model achieve better outcomes.

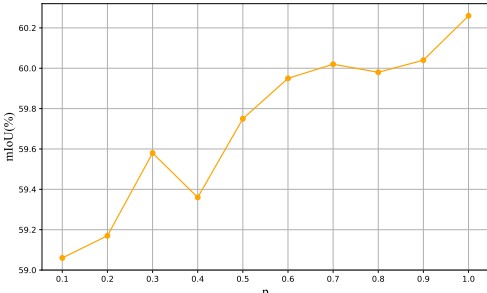

**Figure 6: Effectiveness of noise ratio in the GSM**

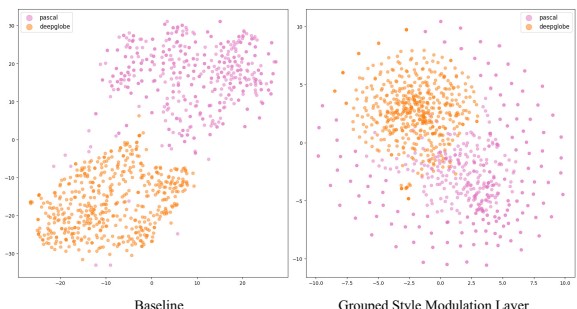

| Baseline | Grouped Style Modulation Layer |

**Figure 7: Visualization results of features on t-SNE**

## 5 CONCLUSIONS

In this work, we introduce a cross-domain few-shot semantic segmentation network based on Dual-Branch Fusion with Style Modulation. By constructing latent style estimations to characterize the distribution range of feature statistics, we explore the potential semantic directions and strengths of features and employ affine transformations to synthesize new feature statistics for domain style enhancement. This approach facilitates the model's learning of diverse domain styles, thereby improving its generalizability. Moreover, to further optimize the prediction masks of query samples, during the testing phase, we employ a dual-branch fusion technique to fine-tune the model. The goal is to enhance the model's ability to adapt to the data distribution of the target domain. Extensive experiments were conducted on the FSS-1000, ISIC, Chest X-ray, and Deepglobe datasets. The results demonstrate that the proposed method is highly competitive in cross-domain few-shot semantic segmentation tasks.

## 6 ACKNOWLEDGMENTS

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
