# OpenReview forum: "Dual-Branch Fusion with Style Modulation for Cross-Domain Few-Shot Semantic Segmentation"
_acmmm.org/ACMMM/2024/Conference — MM2024 Poster_

### Official Review · Reviewer_8heP · 2024-05-09

**Rating:** 2
**Confidence:** 4

**Summary:**

The submitted paper, "Dual-Branch Fusion with Style Modulation for Cross-Domain Few-Shot Semantic Segmentation," addresses the critical challenge of cross-domain few-shot semantic segmentation (CD-FSS), which is highly significant given the practical limitations in labeled data across diverse domains.

**Strengths:**

The paper introduces a Dual-Branch Fusion with Style Modulation (DFSM) that uniquely combines style modulation and dual-branch fusion to tackle domain discrepancies and limited data issues.
1. Grouped Style Modulation (GSM): This parameter-free approach dynamically adjusts feature distributions to enhance model generalizability across domains.
2. Dual-Branch Fusion (DBF): The strategy integrates two branches of modeling to refine the discriminative power of the network, allowing for more accurate and robust segmentation.
3. Comprehensive Evaluation: Extensive experiments on multiple benchmarks such as FSS-1000, ISIC, Chest X-Ray, and Deepglobe, where it outperforms existing methods, showcasing the practical effectiveness and adaptability of the proposed method. Especially in Figure 7, the distribution between the two domains is shown clearly. However, some test tables are described not clearly such as whether the task was 1-shot or 5-shot, and the distribution is still a barrier across different domains, not well integrating into similar distributions.
4. High performance: Can you explain why the model has improved a lot on the ChestX data set, but the improvement on other data sets is not obvious? What is the reason for this?

**Limitations:**

1. Lack of Theoretical Analysis: The paper primarily focuses on empirical results without substantial theoretical backing for the effectiveness of the proposed methods. Especially for the parameter-free approach: GSM, please use some quantitative methods to explain the reduction in the number of parameters.
2. GSM Methodology: It is not clear how the styles are selected and how their optimal relevance to the target domain is determined.
3. Impact of DBF Complexity: The dual-branch architecture increases the model complexity, which might not be justifiable unless a clear benefit is shown over simpler methods. Besides, the two-stage methods, in the fine-tuning stage, the target is used on Dual-branch fusion, whether the target data set are used for model fine-tuning? How many target data sets are used for fine-tuning? Is this unfair to CD-FSS tasks?
4. Generalization over Datasets: While performance improvements are noted, the generalization capability over more diverse or unseen datasets is not discussed, which is important in the CD-FSS field.
5. Parameter Tuning and Scalability: Details on the sensitivity of the model to hyperparameter settings and its scalability to larger datasets are lacking.
6. Lack of novelty: what is the main difference between your work and paper: 'Yang, Y., Chen, Q., & Liu, Q. (2023). A dual-channel network for cross-domain one-shot semantic segmentation via adversarial learning. Knowledge-Based Systems, 275, 110698.' ?
7. Lack of detail comparison with state-of-the-art methods, such as 'Huang, X., Zhu, C., & Chen, W. (2023). Restnet: Boosting cross-domain few-shot segmentation with residual transformation network. arXiv preprint arXiv:2308.13469.'.

**Suitability:**

2

---

### Official Review · Reviewer_GHXk · 2024-05-22

**Rating:** 5
**Confidence:** 2

**Summary:**

This work aims to resolve Cross-Domain Few-Shot Semantic Segmentation task, which mainly faces two challenges: 1) domain gap between source and target domains, 2) limited number of samples in the target samples. To address these two issues, the author proposed an efficient parameter-free Grouped Style Modulation (GSM) layer that generates various styles for intermediate features by utilizing their statistical property within a mini-batch. Then, to tackle the second challenge, they proposed to augment samples from support set as pseudo-support samples and true support-set samples as pseudo-query samples.

**Strengths:**

1. The primary motivation is reasonable and it is well-addressed by appropriate and interesting modules.
2. Extensive experiments demonstrated the effectiveness of the proposed methods, showcasing the promising performance gains.
3. The presentation of the paper is highly qualified.

**Limitations:**

1. Compared with [19], the novelty of GSM is limited, although applying it for style modulation makes sense. Please carefully compare the proposed GSM with [19].
2. The motivation for Dual-Branch Fusion (DBF) strategy is unclear. The author claimed that “the branch with GSM captures excessive class-irrelevant information, the branch without GSM loses some class-relevant information, while the fusion method achieves accurate pixel-level prediction results”, which makes me confused. Why does GSM capture class-irrelevant information? and why does w/o GSM also lose class-relevant information? Isn’t class-relevant important for semantic segmentation? Then how can each branch help each other?
3. The details of augmentation are missing. How can it ensure the augmentation is meaningful? which may significantly affect performance.

[19] Xiaotong Li, Yongxing Dai, Yixiao Ge, Jun Liu, Ying Shan, and LINGYU DUAN. 2022. Uncertainty Modeling for Out-of-Distribution Generalization. In International Conference on Learning Representations.

**Suitability:**

3

---

### Official Review · Reviewer_1UGM · 2024-05-24

**Rating:** 3
**Confidence:** 2

**Summary:**

The paper proposes a Dual-Branch Fusion with Style Modulation (DFSM) method for Cross-Domain Few-Shot Semantic Segmentation (CD-FSS). The DFSM method includes a Grouped Style Modulation (GSM) layer to adapt to feature distribution changes across domains and a Dual-Branch Fusion (DBF) strategy to improve pixel-level prediction accuracy. The approach is designed to be parameter-free, enhancing generalization and discrimination abilities with limited labeled samples.

**Strengths:**

1. a lot of comparison experiments of other domain adaptation work.
2. Includes results of visualization of the features, visualizing the contribution of the proposed module
3. The problem are defined clearly
4. Many visual results are given in the main paper and the supplementary materials to visualize the model performance

**Limitations:**

- Limited Contribution: it seems to be an incremental work of previous work[17] which combine the work [17] and a statical distribution shift module.

 - In section2.1, you need to clarify why you solve this problem by few shot domain adaptation instead of using domain generalization. According to your clarification, in the practical applications scenario, it seems more likely that the information about the target domain may not be available, and it seems that domain generalization is a more robust generalization scheme. It would be better if you could illustrate more advantage of DA compared to DG.

 - As you mentioned in section 2.3, there are also works that use learning methods to bring the target domain closer to the source domain in latent space. It would be beneficial to include comparisons with such methods in your experiments to demonstrate that your method does not degrade performance while maintaining model complexity. Since based on your mission scenario settings, you are not extremely constrained by computational resources.

 -  Line398-401 has some elaboration errors and formatting errors

 - The example you provided in Fig. 5 is somewhat simplistic, which may not fully capture the migration performance of your method in the context of semantic segmentation in remote sensing images. It would be beneficial to examine results on more granular datasets such as SpaceNet, loveDA.

**Suitability:**

2

---

### Meta-Review · Area_Chair_CVFj · 2024-06-30

**Recommendation:** Accept (Poster)
**Confidence:** 3

**Metareview:**

Most reviewers appreciate the innovative combination of techniques proposed in this work. However, one reviewer expresses concerns about the justification of the proposed architectural choices and the demonstration of runtime and memory efficiency. This reviewer expected the authors to provide a comprehensive table presenting quantitative evaluations across various ablation studies and comparisons to state-of-the-art methods.
After reviewing the comments from all four reviewers, I suggest accepting this work. The authors are encouraged to refine the manuscript before submitting the final version.